# Impacts of Environmental Pollution on Brain Tumorigenesis

**DOI:** 10.3390/ijms24055045

**Published:** 2023-03-06

**Authors:** Cristina Pagano, Giovanna Navarra, Laura Coppola, Beatrice Savarese, Giorgio Avilia, Antonella Giarra, Giovanni Pagano, Alessandra Marano, Marco Trifuoggi, Maurizio Bifulco, Chiara Laezza

**Affiliations:** 1Department of Molecular Medicine and Medical Biotechnology, University of Naples “Federico II”, 80131 Naples, Italy; 2Department of Chemical Sciences, University of Naples “Federico II”, 80126 Naples, Italy; 3Institute of Endocrinology and Experimental Oncology (IEOS), National Research Council (CNR), 80131 Naples, Italy

**Keywords:** brain tumor, environmental risk factor, environmental pollutants, heavy metals, toxics

## Abstract

Pollutants consist of several components, known as direct or indirect mutagens, that can be associated with the risk of tumorigenesis. The increased incidence of brain tumors, observed more frequently in industrialized countries, has generated a deeper interest in examining different pollutants that could be found in food, air, or water supply. These compounds, due to their chemical nature, alter the activity of biological molecules naturally found in the body. The bioaccumulation leads to harmful effects for humans, increasing the risk of the onset of several pathologies, including cancer. Environmental components often combine with other risk factors, such as the individual genetic component, which increases the chance of developing cancer. The objective of this review is to discuss the impact of environmental carcinogens on modulating the risk of brain tumorigenesis, focusing our attention on certain categories of pollutants and their sources.

## 1. Introduction

Brain tumors are a group of neoplasms affecting the central nervous system (CNS). They are categorized as primary or secondary tumors, depending on whether they originate directly from the nervous tissue or emerge due to malignancies located outside the cranium that metastasize to the brain. The most prevalent brain tumors are intracranial metastases, meningiomas, and gliomas, specifically glioblastoma [1].

Results from large-scale cohort- and population-based studies on combined CNS tumors or on gliomas alone have shown an increasing incidence of brain tumorigenesis in Northern Europe [2] and, overall, in the industrialized countries, where the rate of new cases of nervous system cancer has been found to be increased. Indeed, the incidence rate of all malignant and non-malignant CNS tumors was 24.25 per 100,000 between 2014 and 2018 in the United States according to the Central Brain Tumor Registry of the United States (CBTRUS). The incidence rate of malignant CNS tumors was 7.06 per 100,000 [3]. The increase in cancer burden due to environmental factors is indeed connected to the degree of industrialization of a country. A greater urbanization rate, a greater number of adult population at risk due to the exposure to certain lifestyles and a greater number of workers assiduously exposed to environmental carcinogens constitute significant cancer risk factors [4].

It is well known that genetic damage, inherited by the cellular progeny, is the basis of neoplastic transformation. Substances that promotes cancer are known as carcinogens and can act directly or indirectly on DNA, causing mutations [5]. Indeed, the carcinogens are classified into two classes: genotoxic and non-genotoxic carcinogens. Genotoxic carcinogens bind directly to DNA, causing mutation in the genetic material. They are dangerous, for which the carcinogenicity does not have margin of tolerance as they represent a cancer risk to humans, even at very low doses. In contrast, non-genotoxic cancinogens affect fundamental processes regulated by or dependent on DNA and gene expression, such as growth and differentiation. For these molecules there is a margin of safety, thus, their use is tolerated unless the exposure or intake level would exceed the threshold values [6]. Non-genotoxic carcinogens have been shown to act as tumor promoters (such as 1,4-dichlorobenzene) or as inducers of inflammatory responses (metals such as arsenic and beryllium) [7]. Environmental carcinogens are so defined because they are chemicals present in the environment that can be absorbed by the human body via food, drink or air. Most importantly, they have been shown to cause cancer in humans and/or in vivo models [8]. The production and subsequent accumulation of environmental carcinogens is the result of human interaction with their surroundings, leading to changes in energy patterns, radiation levels, chemical, physical and biological alterations [9]. The limit values of substances which are dispersed in the environment and that can accumulate in our body are particularly important. For example, arsenic trioxide may be used as a medication to treat a type of cancer known as acute promyelocytic leukemia; 0.15 mg/kg/day daily is the therapeutic dose used, and exceeding this amount has fatal conseqeucnes [10].

One of the biggest environmental threats to human health is air pollution. The air quality guidelines (AQG) present specific recommendations on the levels of the so-called six “classical pollutants”—particular matter (PM10), fine particulate matter (PM2.5), ozone (O₃), nitrogen dioxide (NO₂), sulfur dioxide (SO₂) and carbon monoxide (CO). Outdoor air pollution and PM from outdoor air pollution have been catalogued by the International Agency for Research on Cancer (IARC) as carcinogenic to humans (IARC Group 1) based on sufficient evidence of carcinogenicity in humans and experimental animals and on strong mechanistic evidence [11].

The 2021 guidelines update reflects far-reaching evidence that shows how air pollution affects many aspects of health, even at low levels. The updated recommended guideline levels for the common air pollutants are: PM2.5 should not exceed 5 µg/m^3^, while 24 h average exposures should not exceed 15 µg/m^3^ on more than 3–4 days per year; PM10 (particulate matter with a diameter of 10 microns or less) should have concentrations of 15 µg/m^3^ as an annual mean and 45 µg/m^3^ as a 24 h mean; O_3_ concentrations should be at 100 µg/m^3^ as an 8 h mean; there should be NO_2_ concentrations of 10 µg/m^3^ as an annual average and 25 µg/m^3^ as a 24 h mean; SO_2_ concentrations should be at 40 µg/m^3^ as a 24 h mean and CO concentrations of 7 µg/m^3^ as a 24 h mean [12].

Among these parameters, particulate matter has been catalogued by the International Agency for Research on Cancer (IARC) as among the carcinogenic compounds of Group 1 [13] and their role in brain tumorigenesis is still under study. Further, an association between long-term exposure to PM2.5—which is typical of traffic-related air pollution—and malignant brain tumors has been observed [14].

A different but linked class of plausible factors in tumor risk is water pollution, which is already associated with neurological alteration and diseases [15]. The elevated content of some substances such as nitrites and nitrates in potable water can be detrimental to human health due to the formation of nitrosamine compounds which are considered to be carcinogenic [16]. Furthermore, it is not only the presence of elements foreign to normal sources of drinking water that poses a risk to human health. The conventional water treatment processes used to protect water safety from microorganisms can induce the formation of disinfection by-products such as trihalomethanes (THMs), which have been described as genotoxic and mutagenic [17].

Another class of pollutant that has been known to be a risk factor for brain tumors is ionizing radiation (RAD). RAD, due to its elevated energy, is able to damage DNA. Although exposure to high quantities of RAD has been extensively studied and certainly represents a source of tumorigenesis, the long-term health effects from protracted exposures at low doses are concerning [18]. Regardless, RAD has been associated with an overall risk of meningioma and an increased risk of glioma for younger people [19]. Moreover, there is evidence that occupational exposures lead to a higher risk in brain carcinogenesis. Occupations reported to be associated with brain cancer include electrical and petrochemical workers and farmers whose work was closely associated with pesticide exposure [20]. Environmental exposure to pesticides is specifically worrying for children as they are particularly vulnerable due to physiological and behavioural characteristics. The greater food or fluid intake per body weight and “hand-to-mouth” activity can increase the dose and toxicity in children compared to adults [21].

Among other widespread environmental pollutants, we can find heavy metals such as copper (Cu), arsenic (As), lead (Pb), nickel (Ni), cadmium (Cd) and zinc (Zn), all of which can cause damaging health effects, such as cancers [22]. Morakinyo et al. reported the values obtained for heavy metals compared with the AQG and the USEPA (United States Environmental Protection Agency) regulatory guidelines in their work. Therefore, the admissible limits of concentrations of metals in the air are: Cu 100 µg/m^3^; As 6 ng/m^3^; Pb 0.5 µg/m^3^; Ni 0.24 ng/m^3^; and Cd 0.2 ng/m^3^. It is worth noting that Zn recommended values have not yet been reported in the literature [23].

In addition to these well-known elements, the recent improvement in detection techniques has allowed for the easier identification of a growing number of pollutants and their derivatives which, taken together, can be defined as emerging pollutants (EPs). EPs, also known as contaminants of emerging concern (CECs), have attracted the attention in the scientific community due to their newly recognized potential effects on health or the environment; unfortunately, they do not yet adequate regulation and monitoring [24]. Included in this category we can find the endocrine disruptors (EDs), hormonally active chemicals that have been found to affect the functioning of the endocrine system in animals and humans and, as a result, to cause a wide range of diseases, including brain cancer [25,26].

All organ systems can be targets of toxic exposures. In particular, due to their biological role, the lungs represent one of the organs most in contact with the external environment. The inhalation of air pollution (especially PMs), polycyclic aromatic hydrocarbons (PAHs), and toxic metals has been associated with the development of diseases in the respiratory system, such as asthma, allergies, chronic obstructive pulmonary disease and finally lung cancer [27].

Carcinogens can thus enter the human body through respiration, entering the bloodstream at the level of the alveoli, and through the blood reach and accumulate in various tissues, including the CNS [28].

It is important to highlight that brain tissues are heavily protected by the blood–brain barrier (BBB), which makes the brain an immunologically privileged environment. The endothelial cells (ECs) that form the BBB tightly regulate the movement of molecules, ions, and cells between the blood and the brain [29]. Despite this, some chemical pollutants (BPA, dioxin) have lipophilic characteristics that permit them to cross the BBB, accumulating in the brain parenchyma where they exert harmful effects [30]. Several other substances, such as manganese, iridium, silver, titanium, and cerium dioxides, have been found to accumulate in different brain areas. They trigger inflammation responses, which can be deleterious to the barrier, leading to leakage. The nanoparticles can thus reach the brain without even crossing the BBB [31]. Consecutively, other types of pollutants, such as PCBs, are capable of seriously altering the integrity of the BBB endothelium, leading to a facilitated CNS accumulation [32].

Moreover, the build-up of airborne substances in the brain tissues can happen via olfactory bulb pathways, regardless of absorption in the bloodstream. For instance, ultrafine particles (UFP < 0.1 µm) and PM2.5 has been found to reach the olfactory cortex and other brain regions through the olfactory epithelium and subsequently through the olfactory bulb [33]. Furthermore, PM2.5 can destroy the integrity of the BBB; thus, peripheral systemic inflammation easily crosses the BBB and reaches the CNS [34,35].

On the basis of these notions, which will be explored in depth in our manuscript, we aim to gather the current knowledge of the risks that exposure to different categories of environmental pollutants can cause to human health (Figure 1; Table 1). Understanding the sources of harmful chemicals found in our environment, in conjunction with assessment of their effects on tumorigenesis, can provide researchers with the critical information required to render decisions regarding regulatory initiatives.

## 2. Endocrine Disruptors

Endocrine disruptors (EDs) are chemicals, or chemical mixtures, that interfere with the synthesis and normal functioning of hormones [45]. EDs are comprised of different classes of compounds, such as pesticides, industrial chemicals, plasticizers, nonylphenols, metals, pharmaceutical agents and phytoestrogens. They act at very low doses, have multiple action mechanisms, and comprise an extensive group of substances with different chemical structures. The risk assessment shows that exposure to the substances individually does not constitute a risk, but that the total exposure can be a cause for concern [46]. Some common endocrine disruptors involved in neurotoxicity and brain tumorigenesis are discussed below. Polychlorinated biphenyls (PCBs) are a group of EDs, often used for industrial production as plasticizers, although their production has declined drastically since the 1970s. Characterized by high lipophilicity, they can induce estrogenic effects once they accumulate in the body, especially in fat tissue. Here, in the body, they bind to various receptors, thereby interfering with endocrine-associated pathways, causing a disruption of the body homeostasis [47]. Due to their low degradability rate, environmental PCBs are found in soil and can be released into the air; as such, they have been linked to various toxic effects, prominently the reduction in human fertility. The Occupational Safety and Health Administration (OSHA)’s permissible exposure limit (PEL) is a time-weighted average (TWA) airborne concentration of 1.0 milligrams per cubic meter (mg/m^3^) for PCBs containing 42% chlorine (average molecular formula of C12H7Cl3). The PEL for PCBs with 54% chlorine and an average molecular formula of C12H5Cl5 is 0.5 mg/m^3^ (OSHA 1998) [48].

Recently, there has been growing evidence of PCB neurotoxicity. PCBs are neurotoxicants that have been associated with the disruption of the brain capillary endothelium, leading to a reduction in the functionality of the blood–brain barrier (BBB), thus promoting brain metastasis formation [49]. Furthermore, an epidemiologic study managed to link PCB exposure with neurodegenerative diseases, including Parkinson’s disease, amyotrophic lateral sclerosis, non–Alzheimer-related dementia, and brain cancer in adults [32].

A class of chemicals known as per- and polyfluoroalkyl substances (PFAS) are utilized in a variety of products, including clothes, food containers, and electrical wires [50]. The wide application of these compounds in numerous products, their high environmental persistence, and their long half-life (up to 7 years) makes it for someone to be exposed to them.

The European Food Safety Authority (EFSA)-recommended limit for the PFAS substances is 4.4 ng/kg body weight/week [51].

Once in the body, PFASs can act as EDs, binding to human thyroid hormone transport protein transthyretin (TTR), thus leading to thyroid toxicity, although the precise mechanism underlying this phenomenon is still not known [52]. There is evidence that PFAS can accumulate in the brain [53]. Recently, these factors have caused researchers to concentrate their research on PFASs. In particular, the initial links between PFAS exposure and gliomas, a form of brain tumor, were discovered. It has been found that PFAS can accumulate in gliomas, which is probably thanks to their ability to pass through the BBB and deposit in brain tissue [54].

Significant associations between Perfluorooctanoic acid (PFOA) content, tumor grade and pathogenic molecular markers (such as Ki-67 and P53) were found, indicating that PFAS exposure may be a contributing factor for glioma development [55].

The maximum limit value was set at 1 mg/kg for PFOA and its salts and at 40 mg/kg for PFOA-related compounds [56].

Furthermore, the effects of PFAS on the brain are not limited to this; it has also been seen that PFAS may also influence bone and adipose tissue [57]. Another endocrine disruptor is bisphenol A (BPA), a chemical compound widely used to produce polycarbonate plastics, such as water bottles.

Again, EFSA proposed to reduce the tolerable daily intake of BPA from 4 μg kg of body weight (bw)^−1^ day^−1^ to 0.04 ng kg of body weight (bw)^−1^ day^−1^ [58].

Various studies have shown that these substances may play a role in the pathogenesis of different cancer types such as prostate, ovarian and brain cancer [59]. Exposure to BPA has been shown to increase the growth of breast cancer cells that depend on estrogen availability due to its ability to mimic estrogen. Due to this similarity, there is a positive association between BPA exposure and meningioma, for which estrogens represent a risk factor due to the expression of its receptors (ERα and Erβ) in tumor cells [60]. Indeed, Komarowska et al. showed that both meningioma and glioma are connected with increased concentrations of BPA in patient plasma [61]. Despite these findings, further studies are needed to better associate the exposure of endocrine disruptors to the genesis of brain tumors.

## 3. Air Toxic Pollutants

In most cases, the cause of brain cancer is still largely unknown, but experts say some of the factors that may increase the risk of developing brain cancer include exposure to air toxic pollutants and chemical compounds.

Every year, a large number of people die due to the adverse effects of air pollutants originating from human activities, and their introduction into the environment is due to the large industrialization process that has taken place over the last centuries [62]. Due to their damaging properties, any kind of environmental protection law requires the continuous monitoring of the effects that these chemicals, especially if newly introduced, have on the human body. In this way, it is possible to reduce the incidence of pathologies and to minimize the risks associated with occupational exposure [63]. Mainly, the continuous exposure to toxic pollutants present in the air can cause a wide range of abnormalities in the nervous tissue as much as the onset of malignant brain tumors. These pollutants can enter the body by transmission through the circulatory system and other ways which are still not fully known [28]. Furthermore, air pollution may have a causal role for malignant brain cancer through oxidative stress and neuroinflammation pathways. In fact, the production of circulating cytokines such as TNF α and IL-1β cause neuroinflammation, neurotoxicity and cerebral vascular damage [64].

### 3.1. Pesticides

The term “pesticide” is used to classify a substance or mixture of substances which either prevents, repels, or destroys pests, or which is used as a plant regulator, defoliant, or desiccant [65]. This definition includes, among others, herbicide, insecticide, rodenticide, bactericide, and fungicide. Exposure to pesticides due to occupation has been proposed as a cause of tumorigenesis. This may be due to the composition of these compounds. Referring in particular to farmers, the main workers in the agricultural sector, different studies have suggested the existence of a trend between the prolonged use of these agents and the onset of specific tumor types. This applies for the pesticide atrazine for lung cancer, bladder cancer, non-Hodgkin lymphoma, and multiple myeloma [66]; the herbicide glyphosate for multiple myeloma [67]; and neonicotinoid insecticides for breast cancer [68]. More specifically to the nervous system, there are pesticides that contain alkylureas or amines that metabolize nitroso compounds which have been associated with neurogenic tumors [69]; additionally, some fungicides that contain organochlorides and alkylureas combine with copper sulfates to induce glioblastoma multiforme [70]. We extracted information gleaned from a chosen group of meta-analyses and case–control studies, which we deemed to be convincing based on their size, written over the last few decades to outline whether there is indeed a relationship between pesticide use and increased brain tumor risk.

Going in chronological order, one meta-analysis conducted in the 1998 by Khuder et al. studied 33 peer-reviewed studies, published between 1981 and 1996, finding a risk estimate equal to 1.30 for both genders and concluding that there is a weak association between brain cancer and farming-related chemicals [71]. Ten years later, in 2008, Samanic et al. conducted a hospital-based case–control study on 462 glioma- and 195 meningioma-affected patients whose diagnosis occurred between 1994 and 1998. Globally, the authors found no association between exposure to insecticides and/or herbicides and glioma risk in men or women, while there appeared to be a slight association between meningioma and women with occupational herbicide exposure [72]. More recently, Vienne-Jumeau et al. analyzed results obtained from several case–control studies and some cohort studies in order to observe an effective connection between these two factors. As per their conclusions, the results did not all converge to a single positive link, as associations varied with tumor subtypes, kind of crops, animal farming and even proximity to crops. All in all, the authors recommend proceeding with caution in relation to pesticide exposure [69]. A more recent and large-scale meta-analysis, as of 2021, which collected over 40 years’ of epidemiological literature, concluded that the exposure to chemical pesticides indeed increased the risk of brain cancer, regardless of gender, ethnicity, or source of exposure classification. In their research, Gatto et al. calculated an estimate for farmers exposed to pesticides, suggesting that there could be a dose-dependent response to pesticide contact. This indicates that the longer the exposure period is, the greater the risk of developing cancer will be. Furthermore, differences in pesticide purposes and toxicity seems to be the reason for region-specific estimates found by the group, which appears to be a common denominator in the majority of these studies [73]. In their conclusions, there seems to be a true association between farming and brain cancer. It is also important to note that exposure to pesticides may underlie the development of brain tumors, even in childhood, due to parental occupational exposure. In 2016, a hospital-based case–control study conducted by Chen et al. evaluated the risk of childhood brain tumor with exposure to pyrethroids, a neurotoxic class of synthetic pesticides which have been connected to adverse child neurodevelopment [74,75]. The group recruited 161 affected cases and 170 controls between 2012 and 2015, resulting in increased urinary metabolite levels of pyrethroids and, consequently, increased risk of brain tumor. Despite these conclusions, the group recommend a larger cohort study to better identify the association between this specific type of pesticide and brain tumor [74]. Finally, a meta-analysis conducted by Kunkle et al., performed on 15 epidemiological studies, highlighted the connection between in utero exposure and the development of brain cancer in children [76]. With a risk estimate of 1.48, they found an increased risk of cancer among children whose mothers were exposed to agricultural pesticides, while the risk estimate of paternal exposure, which seems to be as important, especially during the preconception period, was 1.63. Despite the interest shown in this subject and the collective results of these analyses, there is still no clear and confident positive association found between pesticide usage and brain tumors. As suggested by many of the cited articles, it may be advisable for a reduction in the time of exposure, a better protection during their application and a change in the chemicals used for pest control, with a movement instead towards the use of organic pesticides [73]. Furthermore, both parents should avoid pesticide exposure during the time of gestation to decrease any possible long-term health effect on the child.

### 3.2. Heavy Metals

One of the major topics to be investigated in brain tumor development mechanisms is the involvement of trace elements and heavy metals in pathogenetic processes that trigger cancer development and expansion. As is already well known, trace elements are essential chemical elements that play primary roles within the cell, such as stabilizers and enzyme cofactors. A higher concentration of trace elements would thus confer toxic, effects leading to reactive oxygen species (ROS) formation [77]. A more challenging problem in this field is the need to give heavy metals a comprehensive definition: these metals are naturally occurring in the surrounding environment, displaying a high density or relative atomic weight. Some authors have already reported that these metals are involved in the development of different types of cancer given their different concentrations between cancerous and noncancerous tissue. However, it is important to underline that the major limitation, when it comes to brain cancer, is the low availability of healthy tissue as a control due to the inability of scientists to remove adjacent tissue during surgery. This represents a challenge in understanding trace elements and heavy-metal concentration changes in brain cancerous tissue [78]. Nevertheless, research has provided evidence that prolonged exposure to high concentrations of heavy metals like copper, arsenic, lead, nickel, cadmium and zinc could be averse to human health [79]. The aim here is to investigate the presence of heavy metals in different kind of brain tumor cells and outline the connection between these inorganic elements and cancer development. Despite the fact they do not have any biological role in human body, these metals act as pseudo-elements, interfering with physiological metabolic processes in a dose-dependent mode. It is notable how various disorders, such as cancer, spring from excessive damage caused by oxidative stress due to ROS formation.

The involvement of ROS in brain cancerogenesis is currently being actively studied: previous studies have emphasized that enzyme-like malonyl dialdehyde (MDA) level, superoxide dismutase (SOD), and catalase (CAT) activities results were altered in brain tumor tissue [80]. At this stage of understanding, it is possible to believe that mitochondrial antioxidant enzymes may have a pivotal role in the origin of oxidative stress in patients with malignant gliomas. Prolonged exposure to ROS contributes to DNA damage and cancer [81]. A large number of existing studies in the broader literature have examined the contribution of lead in cancerogenesis as it is involved in the oxidative damage process and in the inhibition of DNA synthesis and repair [82,83].

### 3.3. Chemicals Pollutants

Air pollution raises growing concerns about the potential impact of emissions on human health and in particular on the brain. Air pollutants may reach the brain via blood and cross the blood–brain barrier. Benzene and nitrogen dioxide (NO_2_) are commonly used as markers of traffic-related air pollution. Recently, researchers have described an association between brain cancer risk in men and exposure to benzene. The authors have observed that, in men, the benzene represents a stronger risk never-smokers, an intermediate risk in former smokers, and a weaker risk in current smokers. Benzene exposures have been linked to specific childhood brain tumors. Another interesting piece of evidence is the sex difference in benzene-induced toxicities in animal models that may be related to hormone levels. Mice which had been exposed to diesel showed increased neuroinflammation and lipid peroxidation in the brain, alterations that were more severe in male than female mice [84]. Other authors in a population-based case–control study in California have suggested that in utero and infancy exposures to air pollutants generated by industrial activity and road traffic may increase the risk of primitive neuroectodermal tumor (PNET), medulloblastoma, and astrocytoma before 6 years of age. They found a higher risk of embryonic brain tumors in children exposed during brain development from the fetal stage to the first year of life. PNET risks were associated with pre- and postnatal exposure to several toxic substances (butadiene, acetaldehyde, chloroform, perchlorethylene, trichloroethylene), and to first-year exposure of ortho-dichlorobenzene. Medulloblastoma risks were associated with higher rates of exposure to prenatal polycyclic aromatic hydrocarbons (PAHs). For astrocytoma, an imprecise risk was estimated in relation to exposures to lead and some PAHs in the first year of life [85].

### 3.4. Particulate Matter (PM)

Global air pollution exposes more than 50% of the population to toxic air pollutants in the form of particulate matter (PM). PMs are particles, suspended in the air, that include hazardous, non-hazardous, organic and inorganic particles and differ in size, composition, and origin. Furthermore, sources of PM can be both natural and man-made such as the combustion of fossil fuels in vehicles, dust on roads, power plants and various industrial processes. They generally enter the body through the lungs but translocate to essentially all organs. With the increase in industrialization and smog, exposure to PMs has increased significantly and has been determined to be carcinogenic [86]. PMs differ in size and are categorized by the United States Environmental Protection Agency (USEPA) into three classes: (1) coarse particles or PM10, particles with aerodynamic diameter of 10–2.5 µm that can easily penetrate the lungs and get into the blood stream; (2) fine particles or PM2.5, particles with aerodynamic diameter between 2.5 and 0.1 µm, capable of crossing the blood–brain barrier; (3) ultrafine particles or PM0.1, with aerodynamic diameters of 0.1 µm or less, known to cause high oxidative stress. In 2013, the International Agency for Research on Cancer (IARC) catalogued PMs as a carcinogen for humans in Group 1; however, the specific role of PM as tumor initiator or tumor promoter has been widely debated [87].

Harbo Poulsen et al., in a nationwide study of brain tumors in Denmark, showed associations between carbon particles in the air and risk of malignant tumors of the brain, such as malignant gliomas [88]. Some studies identified a positive association between the absorbance of PMs as a result of traffic exposure and the occurrence of brain tumors [14,89]. Mukherjee reported that PMs have been associated with long non-coding RNA (lncRNA) dysregulation, a factor that increases patient predisposition towards the onset or progression of cancer. In particular, the dysregulation of these lncRNAs further leads to the activation of various oncogenic cellular pathways, resulting in the onset or progression of glioblastoma multiforme (GBM) [86]. Moreover, novel observations highlighted an increased risk of developing malignant brain cancer in men, especially Latino men, living in areas with high levels of ambient benzene, ozone, and PM10 levels as compared to women [84]. Additionally, ultrafine particles (UFPs, <0.1 µm) are positively associated with brain tumor incidence and these pollutants may represent a previously unrecognized risk factor for brain tumors [90].

## 4. Water Pollution

Some authors have estimated an association between tap water, exposure to trihalomethanes (THM) and nitrates and neuroepithelial brain tumor risk in young people. The disinfection of drinking water is a common practice to inactivate microbial contaminants and prevent the transmission of infectious diseases. Disinfectants, such as chlorine added to water containing organic matter, result in the formation of disinfection by-products (DBP). Among the most common disinfectants are THMs (World Health Organization, 2017). Another prevalent contaminant in drinking water is nitrate, which forms nitrite via endogenous nitrosation. They are classified as potential human carcinogens. In a study conducted in Iowa (USA), researchers have observed a high risk of developing brain tumor in males who have lived for over 40 years in residences with chlorinated surface water sources compared to those who have not. Another study showed a possible association between the exposure of mothers during pregnancy to DBPs and neurodevelopment in the offspring. This evidence may suggest that certain classes of chemicals found in water are capable of crossing the placenta.

Others results, obtained from the analyses of prenatal and during-the-first-two-years-of-life exposure are not consistent with this hypothesis, and no risk of brain tumor was observed with exposure to THMs [91]. Data have reported that nitroso-compound exposures can cause a base-mispairing action in DNA and mutations in rats and, consequently, permanent cell proliferation [92]. In a study conducted in California, researchers observed neural tube defects in the offspring of women exposed to public water containing high concentrations of nitrates during pregnancy. Other studies have found an interesting dose–exposure relationship between average nitrate levels and neuroepithelial brain tumors. One case–control study conducted in seven countries has investigated prenatal and early postnatal exposure to residential nitrate and nitrite levels in tap water, a value it measured by dipstick. The study failed to point out a direct association between nitrates and brain tumor risk, but highlighted a statistically significant association for nitrite levels greater than 5 mg/L versus undetected levels [93]. Similarly, a previous report showed a significantly increased brain tumor risk for children with detectable nitrite levels in tap water compared to non-detectable levels based on 13 cases and 3 controls, while detectable nitrate levels were not associated with brain tumor risk [94].

Furthermore, another case–control study has evaluated the association between nitrate-nitrogen in public water and childhood malignant brain tumors. The authors observed a statistically significant association for subjects who lived in towns with levels higher than 0.31 mg/L of nitrate-nitrogen in water (>1.37 mg/L nitrate) compared to residents of areas with lower levels. Suggestive results have described that many dietary factors can inhibit nitrosation, including vitamins C and E, as well as substances contained in some fruits and vegetables. Such compounds may reduce or nullify the effects of drinking nitrated water among individuals with a diet rich in these elements. The fetal brain may be susceptible to tumorigenesis, due to the rapid division of neural cells and/or a decreased ability to repair alkylation-induced DNA damage that occurs after nitrate exposure. Thus, prenatal dietary and drinking water effects might be important for childhood health [94].

## 5. Others Environmental Risk Factors

### 5.1. Ionizing Radiation

The etiology of malignant brain cancer remains largely unknown. However, there is growing evidence that ionizing radiation (RAD) and radio frequency (RF) electromagnetic waves are factors involved in the onset of brain tumors [95]. Currently, the International Agency for Research on Cancer (IARC) has classified only RAD as a confirmed carcinogen, with other possible factors still under study [88]. It is important to emphasize that dose limits of RAD do not form a dividing line between hazardous and harmless radiation exposure. Rather, exceeding a limit value means that the likelihood of health effects (particularly cancer) exceeds a defined acceptable value. Since there is no dose value below which it is possible to exclude a risk to health due to ionizing radiation, there is also a certain risk below the limit values which increases as the dose increases. Therefore, any exposure to radiation, even at levels below the specified limit values, should be avoided if possible, and otherwise minimized. The limit value for the effective dose aimed at protecting members of the public is 1 millisievert in a calendar year (Radiation Protection Ordinance). This value refers to all radiation exposures to which members of the public may be exposed due to nuclear and other facilities for the generation of ionizing radiation and the handling of radioactive substances [96].

RAD could cause cancer due to direct damage within cells, in that it specifically could break chemical bonds in DNA. The role of RAD as a risk factor is well established in gliomas, meningiomas, and nerve sheath tumors, particularly in patients who have undergone brain high-dose radiotherapy for cancer treatment in childhood [69]. Moreover, computed tomography (CT) scanning, a diagnostic imaging procedure that use X-rays, exposes the brain to radiation doses. In fact, recent follow-up studies of large cohorts of children and adolescents report an increase the relative risk of developing a brain tumor in such cases [69]. Thus, brain irradiation in childhood even at low doses constitutes a well-established risk of developing the tumor. However, epidemiological studies are fewer and less established in adults. Fortunately, RAD is not a normal environmental factor, compared to RF.

### 5.2. Radiofrequency Electromagnetic Waves

Radiofrequency radiation (RF) is a form of non-ionizing radiation which does not have enough energy to cause cancer by directly damaging the DNA. RF can be emitted by diverse instruments such as mobile phones, radio and television transmissions, wireless networks such as Wi-Fi, satellite communications, microwave ovens or radar [69]. The exposure to RF is widespread due to the overwhelming increase in cellphone use, and since cellphones are usually held close to the head, exposure has been concentrated in the this area. The emitted RF waves cannot directly damage DNA or heat the body tissues, compared to RAD; thus, the precise mechanism through which they can cause cancer remains to be elucidated [97]. A few studies have been designed to focus on prognosis for patients with gliomas, depending upon cellphone use. A study conducted by Melnick et al. on glioma found lower survival in patients with glioblastoma associated with long-term use of wireless phones, while other studies report that RF may cause oxidative damage by inducing an increase in lipid peroxidation and oxidative DNA damage formation in rat frontal lobes [98,99]. Most of the studies conducted on the correlation between brain cancer and RF have been performed using cellular or animal models and, although these results cannot be applied to humans directly, they still provide useful information on possible future developments [95]. However, the results of these studies need further investigation in order to to better understand RF waves impact on human health.

## 6. Study of Genotoxic and Non-Genotoxic Carcinogens

Considering how many new chemical compounds are continuously generated and spread by industrial processes, it is essential to identify as soon as possible all those substances with a carcinogenic effect. Thus, genotoxic chemicals can be identified through in vitro screening to a good degree of accuracy. In vitro screening for anticancer agents can be performed with molecular target-based biochemical assays or cell-based cytotoxic assays [100].

For example, Lopez-Suarez et al. proposed SH-SY5Y cell line as an in vitro cell model for neurotoxicity deriving from various environmental pollutants, such as pesticides, 2, 3, 7, 8-tetrachlorodibenzo-p-dioxin (TCDD), flame retardants, PFASs, parabens, bisphenols, phthalates, and PAHs [101]. Li et al. investigated the role of particulate matter (PM) in regulating activation of astrocytes. The glial cell strain C6 was cloned from a rat glioma which was induced by N-nitrosomethylurea. C6 cells were exposed to different concentrations of PM, revealing that PM stimulated the expression of inducible nitric oxide synthase (iNOS) as well as the production of IL-1b in a dose- and time-dependent manner [102]. As explained before, metals are a threat to human health by increasing disease risk, and some studies linked their exposure to the alteration of miRNA expression. Although several human populations are exposed to low concentrations of As, Cd and Pb as a mixture, most toxicology research focuses on the individual effects that these metals exert on the human body. Martínez-Pacheco et al. observed that a metal mixture was capable of differentially altering miRNA and mRNA gene expression profiles in murine fibroblasts. There is much evidence regarding how these metals may regulate mRNA expression, possibly through genetic mutation, transcription factors deregulation, and epigenetic events [103].

In vivo tests are effective for establishing dose–response relationships between the chemical and biological reactions. Ljubimova et al. selectively exposed rats to coarse (PM2.5–10: 2.5–10 µm), fine (PM < 2.5: <2.5 µm), or ultrafine particles (UFPM: <0.15 µm). They found that intermediate-length PM2.5–10 upregulated the expression of infammation and cancer biomarkers in their brains [104]. Similarly, Ljubimova et al. have analyzed the effects of PM exposure on brain functions using gene microarray analysis. The brain of exposed animals revealed the upregulation of some inflammation-related genes, as well as of specific genes that play a role in tumor onset. For example, Arc is associated with early brain changes and low-grade tumors, whereas Rac1 is associated with long-term PM exposure and highly aggressive tumors. Upon two-week to three-month exposure to coarse PM, Arc was elevated but declined after 10-month exposure. Rac1 was significantly elevated upon 10-month coarse PM exposure. In summary, exposure to air PM leads to distinct changes in rodent brain gene expression similar to those observed in human brain tumors [105].

One of the few conventional models that helps researchers correctly predict the neurotoxicity of various substances is zebrafish. Zebrafish represents a great tool to systematically test the harmful effects of thousands of compounds, which can be challenging for environmental toxicology due to the high number of carcinogens found in the environment. Because zebrafish are transparent during their early life stages, changes in brain morphology are easy to detect, and can be used for the discovery of tumorigenicity due to chromosomal injury [106]. Bourdineaud et al. used zebrafish tissues to test the modulation of brain mitochondrial respiration after exposure of low environmental doses of heavy metals. The study observed a strong inhibition of mitochondrial respiration after exposure to uranium and nanoparticles, and a strong accumulation of methylmercury in the CNS of fish [107]. Pyriproxyfen, a pesticide used as an antiparasitic, has been identified as a partial endocrine disruptor and has been correlated with microcephaly in mammals. Azevedo et al. found that pyriproxyfen can trigger a cascade of effects in the mitochondria of zebrafish brain, which may be the cause of CNS development problems associated with exposure to this pesticide [108]. The possible link between air pollution and central nervous system tumors is supported by several national studies. In a study conducted by Poulsen et al., over 20,000 samples of intracranial CNS tumors were analyzed, in which some pollutants were found to be associated with brain neoplasms other than glioma [109]. Aslak Harbo Poulsen et al. identified 12,928 diagnosed cases of brain tumor in Denmark. The brain tumors examined were all associated with environmental pollutants in particular carbon black and organic carbon [88]. Jørgensen reported through a study conducted in the Danish Nurse Cohort with 28,731 nurses (age ≥ 44 years), a new, strong and significant association between PM2.5 and total brain tumors in obese women, also exploring for the first time a possible association between air pollution and lifestyle [110]. These are just some of the works reporting the significant impact of environmental pollutants on brain tumor development. Studies are still underway to better understand which pollutants are most risky and closely associated with brain cancer.

Carcinogenic compounds can then be tested using in vitro methods with a good degree of accuracy. Unlike genotoxic compounds, “-omics” approaches are currently preferred for the analysis of non-genotoxic carcinogens. Non-genotoxic carcinogens vary greatly in their modes of action and need modern ‘-omic’ technologies to be correctly examined. High-content technologies can indeed rapidly identify chemicals with similar modes of action to those that have already been flagged as toxic. They can also aid in elucidating the mechanism underlying the toxicity of new non-genotoxic carcinogens thanks to the specific cellular responses elicited by them. Indeed, ‘-omic’ technologies are promising, giving us a useful insight into the carcinogenic potential of chemicals, but require further development to reach full maturity [111].

**Table 1 ijms-24-05045-t001:** **Association between environmental pollutants and brain tumor**. List of pollutants and their corresponding brain tumors. EDs: endocrine disruptors; PCBs: polychlorinated biphenyls; PFAS: polyfluoroalkyl substances; PFOA: perfluorooctanoic acid; BPA: Bisphenol A; THM: trihalomethanes; NO_2_: nitrogen dioxide; PAHs: polycyclic aromatic hydrocarbons; PMs: particulate matters; HM: heavy metal; RAD: ionizing radiation; RF: radiofrequency radiation.

Association between Environmental Pollutants and Brain Tumor		
Pollutants	Tumor	References
EDs: PCBs, PFAS, PFOA, BPA	Meningioma, glioma neuroendocrine tumors	[26,32,47,54].
Pesticide	Glioblastoma multiforme, meningioma	[70,71,74].
THM, nitrate	Neuroepithelial brain tumor, brain tumor	[93].
Benzene, NO_2,_ butadiene, acetaldehyde, chloroform, perchlorethylene, trichloroethylene, PAHs	Neuroectodermal tumor, medulloblastoma, astrocytoma	[85,86].
PMs	Glioma, glioblastoma multiforme	[87,89].
HM: copper, arsenic, lead, nickel, cadmium, zinc	Glioma	[81,82].
RAD	Glioma, meningioma, nerve sheath tumors,	[70,97].
RF	Glioma	[100,101].

## 7. Conclusions

This review describes the association between compounds released into the environment and brain cancer risk, summarizing the pertinent literature regarding the impact of pollution and brain tumorigenesis. Humans are potentially exposed to numerous chemicals pollutants and scientific results suggest that some of these compounds are neurotoxicants, causing the onset of brain tumors. Among the substances generated by industrial activity that may have as much of an impact on brain cancer as continous exposure to pesticides and heavy metals are fine microparticles, such as PM2.5, and tap water with high nitrite concentrations. Ultimately, chemical pollution is a global problem whose consequences are not yet adequately and fully known. This is due to the lack of sufficient information and regulation on the thousands of substances that are continuously released into the environment. Further studies are crucial to understand the potential brain cancer risk associated with environmental carcinogens as well as their biological mechanisms of action. Indeed, the need for research on the subject is greater than ever.

## Figures and Tables

**Figure 1 ijms-24-05045-f001:**
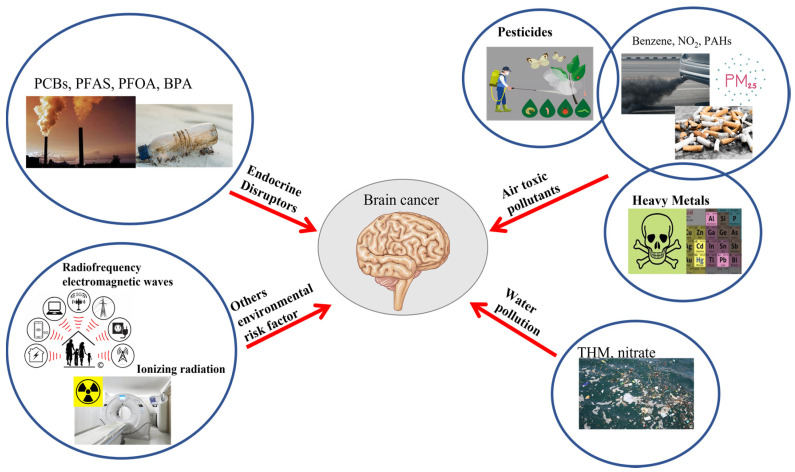
**Schematic representation of the action of different environmental pollutants on brain cancer.** Endocrine disruptors; PCBs: polychlorinated biphenyls; PFAS: polyfluoroalkyl substances; PFOA: perfluorooctanoic acid; BPA: bisphenol A; THM: trihalomethanes; NO_2_: nitrogen dioxide; PAHs: polycyclic aromatic hydrocarbons; PM2.5 µm: particulate matters; Heavy metal; Ionizing radiation; RF: radiofrequency radiation [36,37,38,39,40,41,42,43,44].

## Data Availability

Not applicable.

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
