# Peer review of "Impacts of Environmental Pollution on Brain Tumorigenesis"

_ijms, 2023, doi:10.3390/ijms24055045_

Round 1

Reviewer 1 Report

Extensive English-language editing will be necessary, with some sections requiring more correction than others (esp. plural/singular, verb tense, inappropriate inclusion of 'the' before nouns, etc.). You have many instances of very long sentences (4+ lines of text in the article) - these should be split for ease of reading and clarity.  Avoid using 'familiar' words/phrases (such as 'nowadays').

I strongly recommend using a different abbreviation for Ionizing Radiation (such as RAD). 'IR' is used specifically for 'infrared radiation' - using it as you do is likely to be confusing for your readers.

Unclear why you would include RF and Ionizing Radiation as 'air pollutants' - they should be categorized/treated separately.

 Several sections need revision to harden-up 

Author Response

Dear Reviewer 1,

I am re-submitting the manuscript entitled: “Impacts of environmental pollution on brain tumorigenesis” by Pagano et al. ijms-2146788, for consideration as a Full paper in International Journal of Molecular Sciences.

The manuscript has been revised according to the reviewer's thoughtful comments and uploaded.

Changes should be easily visible to the editors and reviewer since they were clearly highlighted.

Reviewer 1:

Point 1: Extensive English-language editing will be necessary, with some sections requiring more correction than others (esp. plural/singular, verb tense, inappropriate inclusion of 'the' before nouns, etc.). You have many instances of very long sentences (4+ lines of text in the article) - these should be split for ease of reading and clarity.  Avoid using 'familiar' words/phrases (such as 'nowadays').

Reply: An English-language editing was performed by our native speaker colleague. Thanks for the suggestion.

Point 2: I strongly recommend using a different abbreviation for Ionizing Radiation (such as RAD). 'IR' is used specifically for 'infrared radiation' - using it as you do is likely to be confusing for your readers.

Reply: Thank you for the suggestion, we have replaced IR in RAD.

Point 3: Unclear why you would include RF and Ionizing Radiation as 'air pollutants' - they should be categorized/treated separately.

Reply: We have changed “Air Pollutants” in “ Others environmental risk factors”. We believe it may be more suitable to include RF and Ionizing Radiation.

Point 4: Several sections need revision to harden-up

Reply: We have revised all the sections in depth of the paper as suggested.

Reviewer 2 Report

In this review, Pagano et al. discuss potential associations between environmental pollution on brain tumorigenesis. The manuscript is interesting and could be relevant. However, the great flaw of the article is trying to address the effect of multiple types of pollutants on an extremely complex disease. As a result, the discussions presented in each section are brief, superficial, and confusing. Also, there are many specific issues that must be addressed by the authors. Just a few of them are exemplified below:

- Title: The expression "incidence" has a very specific meaning in epidemiological terms that do not fit in the context of this article. I strongly suggest modifying the title to "Impacts of environmental pollution on brain tumorigenesis" or "Associations between environmental pollution and brain tumorigenesis" (or another similar title).

- Abstract, lines 17-20: This excerpt is confusing and poorly written. Please review the coherence and grammar of this excerpt.

- Introduction: The article needs an extensive and professional English review. Some phrases are particularly confusing:  For example: “Some genes regulate the rate of the cell proliferation and/or repair damaged genes as well as genes that should cause the cell death if the damage is beyond repair.”

- Introduction, lines 31-28: Please provide references supporting this information.

- Introduction, general comments: It is not clear whether the authors will address emerging pollutants, classic pollutants, or both. Furthermore, it is not clear whether this is a narrative review or another type of review, nor how the topics were selected and the text is organized.

- Figure 1: Authors must credit the sources of the images used in this figure.

- 2. Endocrine Disruptors, line 97: "Here"? The fat tissue?

- 2. Endocrine Disruptors, line 104: "epidemiologic studies" is mentioned but only one reference is cited at the end of the sentence. Correct it.

­- 2. Endocrine Disruptors, general comment: This section must be divided into paragraphs (in its current form, all section contains a single paragraph).

- 3. Toxic: All the section includes a single short paragraph with a single reference. There are multiple studies that could be cited in this section... Also, brain tumor is not mentioned in this topic even once. I know this topic is used to introduce the other sub-topics, but the authors can work a little more on this introduction. Finally, the term “Toxics” is too generalist and vague.

- 3.1 Pesticides, line 168: "collected in a table the results from several case–control studies". It is not necessary to declare that the data were put in a table.

- 3.2 Heavy Metals and 4. Water Pollution: In both topics, all discussions are made in single paragraphs. Please, divide the text into multiple paragraphs.

- 5. Air pollution: I have not sure if "Ionizing radiation" and "Radiofrequency electromagnetic waves" should be included in the section "Air pollution".

- 5. Conclusions: This conclusion is vague and does not reflect current literature.

Author Response

Dear Reviewer 2,

I am re-submitting the manuscript entitled: “Impacts of environmental pollution on brain tumorigenesis” by Pagano et al. ijms-2146788, for consideration as a Full paper in International Journal of Molecular Sciences.

The manuscript has been revised according to the reviewer's thoughtful comments and uploaded.

Changes should be easily visible to the editors and reviewer since they were clearly highlighted.

Reviewer 2:

Point 1: Title: The expression "incidence" has a very specific meaning in epidemiological terms that do not fit in the context of this article. I strongly suggest modifying the title to "Impacts of environmental pollution on brain tumorigenesis" or "Associations between environmental pollution and brain tumorigenesis" (or another similar title).

Reply: Thank you for the suggestion. We changed the title in “Impacts of environmental pollution on brain tumorigenesis ".

Point 2: - Abstract, lines 17-20: This excerpt is confusing and poorly written. Please review the coherence and grammar of this excerpt.

Reply: The abstract has been revised, thank you for the suggestion

Point 3: Introduction: The article needs an extensive and professional English review. Some phrases are particularly confusing:  For example: “Some genes regulate the rate of the cell proliferation and/or repair damaged genes as well as genes that should cause the cell death if the damage is beyond repair.”

Reply: An English-language editing was performed by our native speaker colleague. Moreover we have eliminated the sentence and remodulated the concept.

Point 4: Introduction, lines 31-28: Please provide references supporting this information

Reply: The introduction has been completely revised and deepened following your suggestions.

Point 5: Introduction, general comments: It is not clear whether the authors will address emerging pollutants, classic pollutants, or both. Furthermore, it is not clear whether this is a narrative review or another type of review, nor how the topics were selected and the text is organized.

Reply: As already reported in the previous comments, the introduction has been completely revised according to your suggestions. We have tried to make the purpose of our manuscript clearer and to make the text more detailed.

Point 6: Figure 1- Authors must credit the sources of the images used in this figure

Reply: it has been done.

Point 7: Endocrine Disruptors, line 97: "Here"? The fat tissue?

Reply: We have made the sentence clearer by remodulating it as follows: “Here, in the body…”

Point 8: Endocrine Disruptors, line 104: "epidemiologic studies" is mentioned but only one reference is cited at the end of the sentence. Correct it

Reply: It has been done

Point 9: Endocrine Disruptors, general comment: This section must be divided into paragraphs (in its current form, all section contains a single paragraph).

Reply: We are not sure that we have understood exactly what you mean by paragraphs, in the meantime we have split the text by inserting a space and the indentation at the beginning of the period. Or do you intend to add new titles to the sections that you have reported to us?

Point 10: Toxic: All the section includes a single short paragraph with a single reference. There are multiple studies that could be cited in this section... Also, brain tumor is not mentioned in this topic even once. I know this topic is used to introduce the other sub-topics, but the authors can work a little more on this introduction. Finally, the term “Toxics” is too generalist and vague.

Reply : As suggested, we have reshaped the text and inserted more references. The term “Toxic” has been replaced with “Air Toxic pollutants”

Point 11: Pesticides, line 168: "collected in a table the results from several case–control studies". It is not necessary to declare that the data were put in a table.

Reply: We have corrected the sentence.

Point 12: Heavy Metals and 4. Water Pollution: In both topics, all discussions are made in single paragraphs. Please, divide the text into multiple paragraphs.

Reply: As mentioned in the previous comment we are not sure that we have understood exactly what you mean by paragraphs, in the meantime we have split the text by inserting a space and the indentation at the beginning of the period. Or do you intend to add new titles to the sections that you have reported to us?

Point 13: Air pollution: I have not sure if "Ionizing radiation" and "Radiofrequency electromagnetic waves" should be included in the section "Air pollution".

Reply: We have changed “Air Pollutants” in “ Others environmental risk factors”. We believe it may be more suitable to include RF and Ionizing Radiation.

Point 14: . Conclusions: This conclusion is vague and does not reflect current literature.

Reply: Thank you for the suggestion, we have revised the Conclusion.

Round 2

Reviewer 1 Report

The paper is still in need of significant English-language editing (despite the authors' assurance that a native-English speaker edited the paper). 

For example

 - line 51: 'they are chemicals derived from the environment' - should say something like: 'they are chemicals that originate in the environment' or 'exposures to these chemicals occur in the ambient environment'

- lines 65- 66: phrases out of order for English language - should be something like "Further, an association between long-term exposure to PM2.5 - which is typical of traffic-related air pollution - and malignant brain tumors has been observed." 

- line 72: 'carcinogenic in a large amount' doesn't make sense in English - do you mean 'at high concentrations'? 'with frequent exposures'? 

- lines 78-79: you are missing a conjunction word somewhere in this sentence

- line 82;  split the very long sentence after the reference on this line: "... exposures at low doses [13]. Regardless, RAD has been . . . "

- line 85 should read: 'Moreover, there is evidence..' 

- line 87 (and repeatedly in the paper): overuse of 'and' in lists. Should only be included before the final item in the list. OR, as seems to be the case here, remove the comma (,) after 'electrical' so it reads "... include electrical and petrochemical workers and farmers . . ."

- lines 88 - 91: long sentence. split into at least 2 (suggest after 'characteristics')

- lines 93-94, why include this statement about pollutants?: 'classified by IARC as both carcinogenic and non-carcinogenic'.  That just means they are among ALL chemicals - if your goal is to indicate that they have been studied and classified by IARC, leave it at that.

- line 104: 'quite ubiquitous' doesn't make sense. If something is 'ubiquitous' it is everywhere. how can it be 'quite everywhere'?

- fold the idea (i.e., the word 'ubiquitous') into the previous sentence. Suggest something like: "... have attracted attention in the scientific community due to their ubiquity and newly recognized..."

-  line 275: 'elements of structure', did you mean 'structural elements' (?)

 - line 283: 'incapacity' is not the correct word here. did you mean 'inability'?

Separately:

 - line 59: AQG does not match words used to make the acronym (Air Quality Directive). Unclear whether the mis-match is due to translation from another language or a typographical error

- Fig 1: random PM2.5 clouds on the diagram aren't clearly associated w/ any of the individual categories noted on the diagram. (2 'clouds' appear to be associated w/ EDs, 1 with Chemical Pollutants?).  Note: EDs ARE chemical pollutants . . . so your categories are ambiguous here. 

- line 133: inconsistent use of EDs vs EDCs

- lines 246-248, what does this mean?: 'possible but of little incidence publication bias'? 

 - line 279: what is a 'quite high density'? 

I have innumerable additional edits but, honestly, I stopped after section 3.3. 

This paper obviously represents a significant body of work, and they authors merit praise for what appears to be a worthy contribution to the scientific knowledge.  However, this manuscript needs considerable work before it is ready for publication. 

Author Response

Dear Reviewer 1,

I am re-submitting the manuscript entitled: “Impacts of environmental pollution on brain tumorigenesis” by Pagano et al. ijms-2146788, for consideration as a Full paper in International Journal of Molecular Sciences.

The manuscript has been revised according to the reviewer's thoughtful comments and uploaded.

Changes should be easily visible to the editors and reviewer since they were clearly highlighted.

We corrected the manuscript according to the revisions requested by the reviewers; Furthermore, we did an additional editing work by modifying various sentences that were not clear and by correcting any English mistakes. The corrections will be visible in track change and highlighted in yellow.

We hope that the editing done by us will be enough since at the moment we don't have the funds to pay for the editing offered by your editorial office and, unfortunately, we don't believe that funds will arrive in the near future.

We apologize for the problem we created, but this is the first time we have been asked to edit English, as we have already published in your magazine several times without encountering this problem.

We would like to say that, in our opinion as much as yours, the topic of the manuscript appears to be very intriguing for the scientific community and that the readability of the text appears to be much more fluid after this editing. Based on these considerations, we hope the manuscript will be considered for publication on IJMS.

Reviewer 1:

The paper is still in need of significant English-language editing (despite the authors' assurance that a native-English speaker edited the paper.

For example: 

Point 1:- line 51: 'they are chemicals derived from the environment' - should say something like: 'they are chemicals that originate in the environment' or 'exposures to these chemicals occur in the ambient environment'

Reply: the line has been replaced with: “Environmental carcinogens are widespread and are so defined because they are chemicals present in the environment that can be absorbed by the human body via food, drink or air and that have been shown to cause, or are suspected to cause, cancer in humans and/or in vivo models”

Point 2:- lines 65- 66: phrases out of order for English language - should be something like "Further, an association between long-term exposure to PM2.5 - which is typical of traffic-related air pollution - and malignant brain tumors has been observed." 

Reply: Thank you, it has been corrected.

Point 3:- line 72: 'carcinogenic in a large amount' doesn't make sense in English - do you mean 'at high concentrations'? 'with frequent exposures'? 

Reply: Thank you, we removed “in a large amount”.

Point 4:- lines 78-79: you are missing a conjunction word somewhere in this sentence

Reply: We split the phrase.

Point 5:- line 82;  split the very long sentence after the reference on this line: "... exposures at low doses [13]. Regardless, RAD has been . . . "

Reply: We split the sentence as suggested.

Point 6:- line 85 should read: 'Moreover, there is evidence..' 

Reply: Thank you, we corrected the mistake.

Point 7: - line 87 (and repeatedly in the paper): overuse of 'and' in lists. Should only be included before the final item in the list. OR, as seems to be the case here, remove the comma (,) after 'electrical' so it reads "... include electrical and petrochemical workers and farmers . . ."

Reply: As suggested, we included an “and” before the final item in the list.

Point 8: - lines 88 - 91: long sentence. split into at least 2 (suggest after 'characteristics')

Reply: We split the sentence in two as suggested.

Point 9: - lines 93-94, why include this statement about pollutants?: 'classified by IARC as both carcinogenic and non-carcinogenic'.  That just means they are among ALL chemicals - if your goal is to indicate that they have been studied and classified by IARC, leave it at that.

Reply: Thank you, we deleted this sentence: “which have been classified by the International Agency for Research on Cancer (IARC) as both carcinogenic and non-carcinogenic”.

Point 10: - line 104: 'quite ubiquitous' doesn't make sense. If something is 'ubiquitous' it is everywhere. how can it be 'quite everywhere'?

Reply: As suggested, we deleted “EPs are quite ubiquitous”.

Point 11: - fold the idea (i.e., the word 'ubiquitous') into the previous sentence. Suggest something like: "... have attracted attention in the scientific community due to their ubiquity and newly recognized..."

Reply: Thank you for the suggestion, we decided to delete the previous phrase.

Point 12:-  line 275: 'elements of structure', did you mean 'structural elements' (?)

Reply: To better clarify the concept, we deleted elements of structure. Thank you.

Point 13:  - line 283: 'incapacity' is not the correct word here. did you mean 'inability'?

Reply: Thank you, we replaced the word with “inability”.

Separately:

Point 14:- line 59: AQG does not match words used to make the acronym (Air Quality Directive). Unclear whether the mis-match is due to translation from another language or a typographical error

Reply: Indeed, AQG stands for Air Quality Guidelines. It has been corrected, thank you.

Point 15: - Fig 1: random PM2.5 clouds on the diagram aren't clearly associated w/ any of the individual categories noted on the diagram. (2 'clouds' appear to be associated w/ EDs, 1 with Chemical Pollutants?).  Note: EDs ARE chemical pollutants . . . so your categories are ambiguous here. 

Reply: As suggested, we remodeled the figure

Point 16:- line 133: inconsistent use of EDs vs EDCs

Reply: Thank you, we corrected the mistake.

Point 17: lines 246-248, what does this mean?: 'possible but of little incidence publication bias'? 

Reply: To make the text clearer, we deleted this sentence: 'possible but of little incidence publication bias';

Point 18: - line 279: what is a 'quite high density'? 

Reply: Thank you, we cancelled the word “quite”.

I have innumerable additional edits but, honestly, I stopped after section 3.3. 

Reply: Thank you, we have edited the subsequent sections, correcting further errors.

This paper obviously represents a significant body of work, and they authors merit praise for what appears to be a worthy contribution to the scientific knowledge.  However, this manuscript needs considerable work before it is ready for publication. 

Reviewer 2 Report

Extensive editing of English language and style is still required.

Author Response

Dear Reviewer 2,

I am re-submitting the manuscript entitled: “Impacts of environmental pollution on brain tumorigenesis” by Pagano et al. ijms-2146788, for consideration as a Full paper in International Journal of Molecular Sciences.

The manuscript has been revised according to the reviewer's thoughtful comments and uploaded.

Changes should be easily visible to the editors and reviewer since they were clearly highlighted.

Reviewer 2:

Point 1: Extensive editing of English language and style is still required.

Reply: We corrected the manuscript according to the revisions requested by the reviewers; Furthermore, we did an additional editing work by modifying various sentences that were not clear and by correcting any English mistakes. The corrections will be visible in track change and highlighted in yellow.

We hope that the editing done by us will be enough since at the moment we don't have the funds to pay for the editing offered by your editorial office and, unfortunately, we don't believe that funds will arrive in the near future.

We apologize for the problem we created, but this is the first time we have been asked to edit English, as we have already published in your magazine several times without encountering this problem.

We would like to say that, in our opinion as much as yours, the topic of the manuscript appears to be very intriguing for the scientific community and that the readability of the text appears to be much more fluid after this editing. Based on these considerations, we hope the manuscript will be considered for publication on IJMS.

Round 3

Reviewer 1 Report

The last several pages need additional editing.  This is the 3rd time I have reviewed this paper, I was hoping that my directions for the first half of the paper would have been carried throughout the paper, but on the first page for which I did not suggest edits (p. 9), I see a single sentence that covers 7 lines of text (!). 

Author Response

Dear Reviewer 1,

I am re-submitting the manuscript entitled: “Impacts of environmental pollution on brain tumorigenesis” by Pagano et al. ijms-2146788, for consideration as a Full paper in International Journal of Molecular Sciences.

The manuscript has been revised according to the reviewer's thoughtful comments and uploaded.

Changes should be easily visible to the editors and reviewer since they were clearly highlighted in green and the track change is active.

Point 1:

The last several pages need additional editing.  This is the 3rd time I have reviewed this paper, I was hoping that my directions for the first half of the paper would have been carried throughout the paper, but on the first page for which I did not suggest edits (p. 9), I see a single sentence that covers 7 lines of text (!).

Reply:

Dear reviewer, we proceeded to re-read the entire text, splitting sentences where they seemed too long to be clear. We have reworded some sentences to make them more understandable. We have applied your suggestions throughout the text where we thought it was needed. We hope we have satisfied you and we thank you for the precise and profound revision you made of the manuscript. This helped us improve the manuscript.

Reviewer 2 Report

I am not satisfied with this manuscript, in terms of content and grammar. After two rounds of review, the manuscript showed limited improvements.

Author Response

Dear Reviewer 2,

I am re-submitting the manuscript entitled: “Impacts of environmental pollution on brain tumorigenesis” by Pagano et al. ijms-2146788, for consideration as a Full paper in International Journal of Molecular Sciences.

The manuscript has been revised according to the reviewer's thoughtful comments and uploaded.

Changes should be easily visible to the editors and reviewer since they were clearly highlighted in green and track change is active.

Point 1:

I am not satisfied with this manuscript, in terms of content and grammar. After two rounds of review, the manuscript showed limited improvements.

Reply:

We are very sorry that you are not satisfied with the grammar and content of our manuscript. We want to say that the manuscript has undergone many changes in the three rounds of revision, thanks to the suggestions of the reviewers. If you look at all the changes made from the first to the last round, you can see how profoundly the text has been changed, both in terms of English and content. We hope that this latest change may have been sufficient to have improved the manuscript.
